# Telomere Length and Telomerase Activity of Granulosa Cells and Follicular Fluid in Women Undergoing In Vitro Fertilization

**DOI:** 10.3390/antiox12020419

**Published:** 2023-02-08

**Authors:** Sándor Péntek, Ákos Várnagy, Bálint Farkas, Péter Mauchart, Krisztina Gödöny, Tímea Varjas, Tamás Kőszegi, Péter Kaltenecker, Rita Jakabfi-Csepregi, Kálmán Kovács, József Bódis, Endre Sulyok

**Affiliations:** 1National Laboratory on Human Reproduction, University of Pécs, 7622 Pécs, Hungary; 2Department of Obstetrics and Gynecology, University of Pecs Medical School, 7624 Pécs, Hungary; 3MTA-PTE Human Reproduction Scientific Research Group, University of Pécs, 7624 Pécs, Hungary; 4Department of Public Health Medicine, Medical School, University of Pécs, 7624 Pécs, Hungary; 5Szentágothai Research Centre, University of Pécs, 7624 Pécs, Hungary; 6Faculty of Health Sciences, University of Pécs, 7621 Pécs, Hungary

**Keywords:** telomere length, telomerase activity, oxidative DNA damage, in vitro fertilization

## Abstract

This study aimed to evaluate the interrelationship between telomere length, telomerase activity and oxidative DNA damage in patients undergoing in vitro fertilization (IVF). This single-center, observational clinical study comprised 102 unselected, consecutive patients with various infertility diagnoses. Granulosa cells (GCs) and follicular fluid (FF) were analyzed simultaneously for telomere functions and for the marker of oxidative DNA damage, 8-hydroxy-2-deoxyguanosine (8-OHdG). An Absolute Human Telomere Lengths Quantification qPCR Assay kit and Telomerase Activity Quantification qPCR Assay kit (Nucleotestbio, Budapest, Hungary), as well as an 8-OHdG ELISA kit (Abbexa Ltd., Cambridge, United Kingdom) were used for analyses. Similar telomere lengths were found in GCs and FF, however telomerase activity was markedly depressed, while 8-OHdG levels were markedly elevated in FF compared with those in GCs (*p* < 0.01). Telomere lengths were independent of telomerase activity both in GCs and FF. However, GC 8-OHdG was inversely related to telomerase activity in GCs and FF (*p* < 0.05). Importantly, 8-OHdG levels both in GCs and FF had significant negative impact on the number of the retrieved and MII oocytes (*p* < 0.01), whereas FF 8-OHdG was negatively related further to the number of fertilized oocytes and blastocysts (*p* < 0.01). In conclusion, we could not confirm the direct association of telomere function and reproductive potential. However, oxidative DNA damage, as mainly reflected by 8-OHdG, adversely affected early markers of IVF outcome and clinical pregnancies.

## 1. Introduction

The importance of telomere dysfunction in reproductive aging has been widely studied, and evidence has been provided for the association of telomere shortening with the failed assisted reproductive treatment [1,2,3,4,5]. Telomeres consist of non-coding guanine-rich tandem repeats of TTAGGG and protein complex. They serve as a cap-like structure to maintain genomic integrity and stability [6,7]. Upon each cell division, telomere length decreases by 50–200 base pairs and without efficiently operating protective mechanisms it progresses to cellular senescence and apoptosis [8,9]. Telomere length can be maintained by the enzyme telomerase, which is abundantly expressed in stem cells, germ cells and regenerating tissues but not in somatic cells; the latter ones, therefore, are not able to compensate for the successive losses of telomere base pairs and undergo telomere shortening [10,11].

Telomerase is a ribonucleoprotein enzyme that adds new telomere sequences to the ends of chromosomes. It is composed of the catalytic unit telomerase reverse transcriptase (TERT) and its RNA template (TERC) [12,13]. These components are associated with six individual proteins of the sheltering complex and several nuclear proteins that function to assist telomere assembly, trafficking and stability [14,15,16]. Mutations or deletion of any components of the telomerase complex may result in telomere shortening, dysfunction, accelerated aging and different clinical manifestations of telomeropathies [14,16,17,18,19,20]. Concerning female reproduction basic studies on folliculogenesis, oogenesis and embryogenesis have revealed the developmental regulation of telomere functions [21,22] and have shown that genetic and/or environmental factors, especially oxidative stress (OS), can compromise their developmental course [23,24,25]. Convincing evidence has been provided for the causal relationship of telomere shortening and the reduced success rate of in vitro fertilization (IVF). Furthermore, it has also been demonstrated that the predictive value of the expression of telomerase for IVF outcome outperforms that of telomere length [26].

The present study was undertaken to evaluate the impact of oxidative DNA damage on telomere functions by measuring simultaneously telomere length, telomerase expression and 8-hydroxy-2′-deoxygnuanosine (8-OHdG) in IVF patients. Attempts were also made to assess the clinical value of follicular fluid (FF) in comparison with granulosa cells (GCs) to analyze OS-related telomere dysfunction and its clinical implications in patients receiving IVF therapy.

## 2. Materials and Methods

### 2.1. Patients, Pretreatment Investigation and Superovulation Protocols

A total of 102 consecutive, non-selected patients who took part in IVF procedures were enrolled in this single-center, observational clinical study in the Assisted Reproduction Unit, Department of Obstetrics and Gynecology, University of Pécs, Hungary, during the period between 1st of January and 3st of May 2022.

The indications for the treatment were the following: male factor (n = 52; 50.98%), tubal occlusion (n = 31; 30.39%), endometriosis (n = 14; 13.73%), advanced maternal age (above 35 years) (n = 10; 9.8%), unexplained infertility (n = 10; 9.8%) and other female causes (n = 7; 6.86%). The exclusion criteria were severe chronic metabolic and autoimmune diseases, known chromosomal and developmental abnormalities of the genital tract, malignant diseases and previously performed chemo- and/or radiotherapy. The clinical characteristics of the patients and the IVF cycles are shown in Table 1.

Before the IVF procedures, the following medical examinations were performed: cervical smear, hysteroscopy or hystero-contrast-sonography (HyCoSy), and serum hormone measurements on the 3rd and 21st day of an unstimulated menstrual cycle within a three months period before the beginning (follicular stimulating hormone (FSH) and luteinizing hormones (LH), prolactin, estradiol, progesterone, testosterone, thyroid-stimulating hormone), and human immune-deficiency virus, hepatitis-B surface antigen and lues screening tests.

The IVF cycles were initiated and managed by three independent physicians. For the superovulation protocol we used a short or long agonist with GnRH-agonist (triptorelin-acetate, GonapeptylTM Ferring Pharmaceuticals, Saint-Prex, Switzerland) or antagonist with GnRH-antagonist (cetrorelix-acetate, CetrotideTM, Merck, Darmstadt, Germany).

### 2.2. Collection of Follicular Fluid and Granulosa Cells

After the follicular fluid (FF) was aspirated from the patients transvaginally with a thin sterile needle, we separated the oocytes from the surrounding granulosa cells (GCs) by G-MOPSTM medium (VitrolifeVR, Goteborg, Sweden) and checked the procedure by transmitted light microscope.

The FF was centrifuged for 15 min at 10,000 rpm, and then the supernatant was divided into two parts for 8-hydroxy-2′-deoxyguanosine (8-OHdG) and for the telomere length and telomerase activity measurements. Both were frozen and kept at −80 °C for further analysis.

After 2–4 h of incubation, we denuded the eggs before ICSI fertilization with hyaluronidase containing HYASE-10XTM medium (VitrolifeVR, Goteborg, Sweden), and divided the remained GC mass into two sections: one for telomere length and telomerase activity measurements (frozen and stored at −80 °C) and one for GC counting and 8-OHdG level analysis. Before the counting, 200 µL of the GC samples were centrifuged at 18 °C (1000× *g*, 10 min) and the pellets were resuspended in 200 µL ice-cold PBS. The washed cells were counted, the process was carried out at 0 °C (cells kept in melting ice) on the day of aspiration, and the samples were stored at −80 °C until the analyses.

### 2.3. Fertilization Methods

In all patients we performed ICSI fertilization alone, or there were several cases where we combined ICSI with the conventional IVF method. The indications for ICSI are mainly semen and sperm-cell abnormalities, advanced maternal age (above 35 years of age) or more than two unsuccessful previously performed conventional IVF treatments. The denuded oocytes before ICSI were placed in G-MOPSTM media (VitrolifeVR, Goteborg, Sweden), and assessed for maturity. For the following breeding, the fertilized oocytes were put in G1TMPLUS or G-TLTM single-step breeding media (VitrolifeVR, Goteborg, Sweden). The conventional IVF method was performed in G-IVFTM PLUS media (VitrolifeVR, Goteborg, Sweden), and 24 h later the fertilization was assessed (presence of two pronuclei cells), and these cells were also placed into the previously mentioned G1TMPLUS or G-TLTM media.

Only Grade 1 staged embryos were allowed to transfer (based on the ESHRE’s Consensus embryo scoring system) 3–5 days after the oocyte retrieval procedure in the cleavage or blastocyst embryo division stage. Embryo transfers (ET) were always controlled by transabdominal ultrasound. One, two or three embryos could be placed in the uterine cavity according to the patient’s wish. The success of the therapy was checked with serum human chorionic gonadotropin beta (beta-hCG) levels 14 days after ET. At this time, we checked for the presence of a biochemical pregnancy, as it is the first-line success indicator of our study.

### 2.4. Laboratory Measurements

The 8-OHdG levels were analyzed both from FF and from GC extracts. The GC samples were thawed at 25 °C, vortexed and centrifuged at 1000× *g* for 10 min at 4 °C. The supernates were used for 8-OHdG measurements (after thawing, washing and centrifugation) by enzyme-linked immunosorbent assay (ELISA) kits by (Abbexa Ltd., Cambridge, UK). The ELISA plates were evaluated at 450 nm in a Biotek Synergy HT reader (Agilent, Santa Clara, CA, USA). This competitive ELISA kit operates with an antibody binding to 8-OHdG with an assay sensitivity of 0.94 ng/mL. Its intra- and inter-assay coefficients of variations were less than 10%.

In the case of GC 8-OHdG analysis, the measured values referred to the number of oocytes. On the day of oocyte retrieval, the concentration/number of GCs was measured with a Countstar BioLab Automated Cell Counter IE 1000 (Alit Biotech, Shanghai, China): 20 µL of each GC suspension was pipetted into a Countstar Chamber Slide, specifically designed for this system, the slide was inserted into the instrument and then the cell number was counted automatically. The 8-OHdG levels in FF were corrected for the number of oocytes.

### 2.5. Telomere Length and Telomerase Activity Analysis

#### 2.5.1. DNA Isolation

The Extractme RNA & DNA Kit (Nucleotestbio, Budapest, Hungary) was used for DNA isolation according to the manufacturer’s protocol.

The GC suspension and the FF were centrifuged in a 1.5 mL falcon tube at 400× *g* and the cells were resuspended in PBS buffer. Then the cells were washed twice with 1 mL cold PBS buffer. The cell suspension was placed in a 2 mL tube. Then, 600 μL Lys Buffer was added into the cell pellet and vortexed for 60 s. Afterward it was centrifuged for 120 s at 15,000× *g*. The supernatant was transferred into a DNA Purification Column and placed in a collection tube and was centrifuged for 30 s at 15,000× *g*. The filtrate was kept, then the DNA Purification Column was placed in a new 2 mL Collection Tube for further DNA purification.

Then, 700 μL DW1 Buffer was added into the DNA Purification Column and was centrifuged for 15 s at 15,000× *g*. The filtrate was discarded, and the collection tube was reused. Then 500 μL W2 Buffer was added and the column was centrifuged for 15 s at 15,000× *g*. The filtrate was discarded, and the collection tube was reused. The collection tube was centrifuged for 90 s at 15,000× *g*. The filtrate was discarded, and the purification mini column was transferred to a sterile nuclease-free 1.5 mL Eppendorf tube carefully. Then 50 μL elution buffer DEB was added and was centrifuged for 60 s at 15,000× *g* to elute purified DNA. The DNA purity and quantity were determined with the MaestroNano Micro-Volume Spectrophotometer (MaestroGen Inc., Hsinchu, Taiwan).

#### 2.5.2. Telomere Length

ScienCell’s Absolute Human Telomere Length Quantification qPCR Assay Kit (AHTLQ) is designed to directly measure the average telomere length of a human cell population. The telomere primer set recognizes and amplifies telomere sequences. The single copy reference (SCR) primer set recognizes and amplifies a 100 bp long region on human chromosome 17 and serves as reference for data normalization. The reference genomic DNA sample with known telomere length serves as a reference for calculating the telomere length of target samples. The Absolute Human Telomere Length Quantification qPCR Assay Kit was used according to the manufacturer’s protocol. For the reference genomic DNA sample, two qPCR reactions were prepared, one with telomere primer stock solution, and one with SCR primer stock solution. Then 20 μL qPCR reactions for one well was prepared as shown below (Table 2).

For each genomic DNA sample, two qPCR reactions were prepared, one with telomere primer stock solution, and one with SCR primer stock solution. Then 20 μL qPCR reactions for one well was prepared as shown below (the instrument used for the qPCR assays was the LightCycler^®^ 480 II (Roche Magyarország Kft, Budapest, Hungary) (Table 3).

For each genomic DNA sample, two qPCR reactions were prepared, one with telomere primer stock solution, and one with SCR primer stock solution. Then 20 μL qPCR reactions for one well were prepared as shown below (Table 4).

Quantification Method: Comparative ΔΔCq (Quantification Cycle Value) Method from manufacturer’s protocol.

#### 2.5.3. Telomerase Activity

ScienCell’s Telomerase Activity Quantification qPCR Assay Kit (TAQ) is designed to quantitatively compare telomerase activity among cell populations. The result was expressed as relative telomerase activity: the telomerase activity of our samples compared with the telomerase activity of the standard sample. The Telomerase Activity Quantification qPCR Assay Kit was used according to the manufacturer’s protocol. The total volume of cell lysis buffer to be used for the samples at 20 μL/million cells was determined. The calculated amount of cell lysis buffer with 5% extra was transferred to a new pre-chilled tube. The exact amount of supplemented cell lysis buffer (0.1 M PMSF and β-mercapto-ethanol) was transferred to each cell pellet sample at 20 μL/million cells. The cell pellet was pipetted carefully up and down 20 times with a 1 mL pipette tip. The homogenized samples were incubated at 4 °C for 30 min. The samples were centrifuged at 15,000× *g* for 20 min at 4 °C. Then 15 μL of supernatant/million cells was transferred to a new pre-chilled tube without disturbing the pellet. Telomerase reactions for each sample were prepared as follows (Table 5).

The total volume of cell lysis buffer to be used for the samples at 20 μL/million cells was determined. The homogenized samples were incubated at 4 °C for 30 min. The samples were centrifuged at 15,000× *g* for 20 min at 4 °C. Then 15 μL of supernatant/million cells was transferred to a new pre-chilled tube. Telomerase reactions were incubated at 37 °C for 3 h. Stopping the reactions was achieved by heating the samples at 85 °C for 10 min. The qPCR reactions and qPCR program setup are summarized as follows (the instrument used for the PCR assays was the LightCycler^®^ 480 II (Roche Magyarország Kft, Budapest, Hungary) (Table 6 and Table 7).

Results Interpretation and Calculations: Comparative ΔΔCq (Quantification Cycle Value) Method from user manual.

### 2.6. Ethical Approval and Consent to Participate

The study was authorized after evaluation by the Regional and Institutional Research-Ethical Committee, University of Pécs, Hungary (8751—PTE 2021). All participants in the study signed the informed consent. The research follows the principles set in the Declaration of Helsinki.

### 2.7. Statistical Analysis

Normality of data distribution was analyzed by the Shapiro–Wilk test. Depending on distribution, the clinical and laboratory parameters were compared by using the Student’s *t*-test or Wilcoxon signed rank test, as appropriate. The association between two continuous variables was assessed by the Spearman’s rank test. Data are expressed as median with minimum and maximum values or as mean ± standard deviation, as appropriate. *p* < 0.05 was considered statistically significant. All analyses were performed in the R statistical environment.

## 3. Results

Table 8 shows telomere length, telomerase RNA expression and 8-OHdG levels in GCs and FF in the whole study population, and separately in those who became pregnant or failed to do so. No discernible differences could be detected in telomere length between GCs and FF, however telomerase RNA expression proved to be markedly depressed (*p* < 0.01), while 8-OHdG was significantly elevated (*p* < 0.0001) in FF compared with GCs when all patients were considered as a single group. When these variables were analyzed separately in the pregnant and non-pregnant groups, they proved to be similar in the two groups. It is of note, however, that 8-OHdG tended to be higher in both GCs and FF in the non-pregnant patients, but its increase did not reach statistical significance.

To analyze further the possible association of telomere length, telomerase RNA component and 8-OHdG with pregnancy, their values were separated into two groups by their median. Accordingly, the percentage of patients with smaller or larger than the median of the respective parameters was compared in the pregnant and non-pregnant groups. As it can be seen in Table 9, no significant differences could be detected in the percentage of patients with a telomere length and telomerase RNA component of less or more than their median values. However, FF 8-OHdG over the median was more frequently seen in the non-pregnant than in the pregnant patients (*p* < 0.05). 

In search of the association between the parameters studied we found a significant direct relationship of telomere length in GCs to that in FF (r = 0.32, *p* < 0.01) but the telomere length in GCs proved to be independent of telomerase RNA expression either in GCs or in FF. Furthermore, there was a strong, positive correlation between 8-OHdG levels in GCs and in FF (r = 0.53, *p* < 0.001). With respect to the impact of 8-OHdG on telomere function, no association was observed between telomere length and 8-OHdG levels measured either in GCs or in FF. However, 8-OHdG in GCs was inversely related to telomerase RNA expression, both in GCs (r = −0.24, *p* < 0.05) and in FF (r = −0.20, *p* = 0.05).

In addition to clinical pregnancy as an outcome measure, several indices of the early phase of fertilization were also evaluated. As presented in Table 10, telomere length and telomerase activity neither in GCs nor in FF were significantly related to germinal vesicles, M I and M II oocytes, fertilized oocytes and blastocysts. At the same time, 8-OHdG both in GCs and FF had a significant negative impact on the number of retrieved and MII oocytes (*p* < 0.01), whereas FF 8-OHdG levels were further negatively related to the number of fertilized oocytes and blastocysts (*p* < 0.01). Other early indices of reproductive potential remained unaffected.

To establish the possible involvement of telomere function in the control of reproductive processes, confounding variables were also considered. Telomere length and telomerase RNA were corrected for maternal age, BMI, estradiol levels and the number of previous IVF treatments. However, even after such adjustments we failed to document significant associations of telomere length/telomerase RNA and the reproductive performance of our patients undergoing IVF.

## 4. Discussion

The present study of IVF patients indicates that telomere length in FF is a reliable measure of telomere length in GCs but it is independent of telomerase RNA, which is markedly depressed in FF compared with in GCs. Importantly, telomere length/telomerase RNA neither in GCs nor in FF can serve as a marker of the success of reproduction. Furthermore, 8-OHdG was found to accumulate in FF and its cumulative FF level was adversely related to telomerase RNA and also to the number of retrieved, M2 and fertilized oocytes and to the formation of blastocysts.

In clinical studies to explore the prognostic value of the telomere functions, various cell types were analyzed in IVF patients with a wide spectrum of infertility diagnoses. In a series of early studies using spare human oocytes, a short telomere has been found to be associated with meiotic dysfunction [27], embryo fragmentation [28] and aneuploidy [29], and with impaired overall pregnancy outcomes even after considering the relevant clinical parameters [30].

Telomere length is maintained and its lengthening is mostly achieved by the telomerase enzyme complex. It has been shown to undergo marked changes during oocyte maturation and following fertilization. Low levels of telomerase activity have been detected in ovulated oocytes followed by a transient increase during the early cleavage cycles after fertilization and a decrease in six-to-eight-cell embryos, morulae and blastocysts [4]. The apparent dissociation of the developmental course of telomere length and telomerase activity suggests that factors other than telomerase are involved in telomere reprogramming for lengthening. With this contention in line, telomerase-null mice have been documented to elongate their telomeres after activation [31].

Ovarian GCs and peripheral leukocytes are widely used to evaluate telomere function in IVF patients. GCs are somatic cells that retain properties of germline stem cells to intensively proliferate. Furthermore, bidirectional communication between oocytes and GCs achieved by gap junctions and paracrine factors ensures nuclear and cytoplasmic maturation of oocytes and their potential for fertilization and embryo development [32,33]. Based on the functional characteristics and critical role of the GCs in regulating oocyte maturation, these cells obtained during IVF are used to assess the reproductive potential of oocytes.

The studies of GCs retrieved from IVF patients have revealed that occult ovarium insufficiency and polycystic ovary syndrome (PCOS) are associated with short telomeres and low telomerase activity [34,35]. Moreover, women with high telomerase activity have higher rates of embryo implantation and clinical pregnancy [36], and telomerase activity has proved to be more significant for predicting IVF outcome than telomere length [26]. Convincing evidence has also been provided that leukocyte telomere length might serve as a reliable biomarker of oocyte quality and reproductive potential [37,38,39]. Interestingly, Wei et al., reported significantly longer telomeres in GCs and peripheral leukocytes collected from PCOS patients [40].

In our present study, both telomere length and telomerase activity in GCs appeared to be independent of outcome measures. The reason for our failure to confirm the predictive value of telomere function is not apparent. However, differences in infertility diagnosis, and genetic- environmental- and life-style factors are to be considered [24,41,42]. It is the major message of our study that samples of FF can replace those of GCs for measuring telomere lengths but not for telomerase activity. Moreover, the 8-OHdG level is markedly elevated in FF, as after its release from the ovarian cells it accumulates in this fluid compartment. Consequently, its cumulative levels appear to reflect oxidative DNA damage better than samples of GCs. With this contention in line, we could demonstrate that some IVF outcome parameters were negatively affected by FF 8-OHdG but not by GCs 8-OHdG.

The role of oxidative stress (OS) in female reproduction has been extensively studied and comprehensively reviewed. It has been clearly demonstrated that excessive generation of reactive oxygen species (ROS) that exceeds the capacity of antioxidant defense mechanisms results in OS, and ROS react with essential cellular elements and compromise oocyte development, fertilization and embryo formation [43,44,45].

It has been established that 8-OHdG is a reliable indicator of oxidative DNA damage and its FF and GC levels have been shown to affect inversely the quality of oocytes and embryos in IVF patients [46,47,48,49]. Investigations of the association of 8-OHdG with telomere functions appears to be particularly relevant because telomeres are composed of short, guanosine-rich tandem repeat sequences that are particularly prone to oxidative damage [50] and the IVF procedure itself has been shown to enhance ROS generation. During IVF therapy, ROS production can be induced externally in the ART setup, such as atmospheric oxygen, centrifugation, consumables, CO2 incubators, temperature, humidity, additions to culture media, freeze and thawing, the IVF-ET technique itself, and visible light, but internally as well from immature oocytes, sperm and embryos [51]. As a combined effect, OS accelerates telomere shortening, decreases telomerase activity and induces senescence/apoptosis via DNA damage-induced activation of the p53 pathway [52].

With respect to some inconsistencies of telomere functions in female reproduction, further large-scale studies are to be conducted to get better insight into the pathophysiological and clinical roles in the telomere system in reproductive processes. Particular attention is to be given to revealing the involvement of epigenetic modifications by DNA methylation/demethylation [4,49] and by histone acetylation/deacetylation cycles [53,54,55] in the control of oocyte maturation and embryo development. The generation of reactive oxygen species and the possible ways to protect the guanine-rich, oxidative stress-sensitive telomeres are also of clinical importance [23,56,57]. It may also be rewarding to investigate the relationship of telomere length to telomerase activity and to the Alternative Lengthening Telomere (ALT) process. This latter is achieved by telomeric R loop-induced DNA damage response that triggers break-induced telomere synthesis to maintain telomere length without telomerase [57].

## 5. Conclusions

In conclusion, in our clinical settings we could not confirm a direct association between telomere function and the reproductive potential in women receiving IVF treatment. However, oxidative DNA damage, reflected by 8-OHdG levels, particularly in FF, adversely affected the early measures of IVF outcome and clinical pregnancy independent of the telomere system.

## 6. Study Limitations

Our study was observational in nature and included relatively few number of patients with heterogeneous infertility diagnoses. It was impossible, therefore, to create subgroups to establish etiology-dependent associations. Furthermore, the complexity of the redox status of GCs and FF was not evaluated by measuring pro- and antioxidant factors, and epigenetic modifications, such as smoking, dietary differences, physical activity, sleeping or stress, so several confounding factors could not be considered. Special effort was made to reveal the role of oxidative damage of telomeric DNA (as indicated by 8-OHdG) because its guanosine rich tandem repeat sequences are particularly sensitive to OS.

## Figures and Tables

**Table 1 antioxidants-12-00419-t001:** Descriptive statistics of the patients studied.

Characteristics	Mean Value	Standard Deviation
Age (years)	35.11	4.89
Body Mass Index (kg/m^2^)	26.08	6.45
No. of previously performed IVF cycles	2.1	1.08
Infertility diagnosis (%)		
Male factor (n = 52)	50.98	
Tubal occlusion (n = 31)	30.39	
Endometriosis (n = 14)	13.73	
Advanced maternal age (n = 10)	9.80	
Unexplained (n = 10)	9.80	
Other female factors (n = 7)	6.86	
Serum estradiol (pmol/L)	1749	1745
Total dose of gonadotropin (IU)	2601	834
Protocol of controlled ovarian stimulation (%)		
Antagonist	57.84	
Short agonist	12.75	
Long agonist	29.41	
No. of retrieved oocytes	10.03	6.34
Metaphase II oocytes—matured	6.52	4.93
Metaphase I oocytes—immature	1.47	1.58
Oocytes with germinal vesicle—immature	1.2	1.76
Fertilization method (%)		
Intracytoplasmatic Sperm Injection—ICSI	100	
Conventional IVF + ICSI	8.91	
Fertilized oocytes with two pronuclei—2PN cell	3.35	3.16
Grade 1 embryos at day 3 (cleavage stage embryo)	3.42	2.71
Grade 1 embryos at day 5 (blastocyst)	2.23	2.3
Chemical pregnancies	27	
hCG mean value in case of chemical pregnancies (IU)	1283	1010
No. of subjects (n)	102	

**Table 2 antioxidants-12-00419-t002:** Items used for the ScienCell’s Absolute Human Telomere Length Quantification qPCR assay.

Reference genomic DNA sample	1 μL
Primer stock solution (Telomere or SCR)	2 μL
2^x^ GoldNStart TaqGreen qPCR master mix (Cat #MB6018a-1)	10 μL
Nuclease-free H_2_O (Cat #8918c)	7 μL
Total volume	20 μL

**Table 3 antioxidants-12-00419-t003:** Items used for the LightCycler^®^ 480 II qPCR assay.

Genomic DNA template	1 μL
Primer stock solution (Telomere or SCR)	2 μL
2^x^ GoldNStart TaqGreen qPCR master mix (Cat #MB6018a-1)	10 μL
Nuclease-free H2O (Cat #8918c)	variable
Total volume	20 μL

**Table 4 antioxidants-12-00419-t004:** qPCR reaction setup.

Step	Temperature	Time	Number of Cycles
Initial denaturation	95 °C	10 min	1
Denaturation	95 °C	20 s	32
Annealing	52 °C	20 s	
Extension	72 °C	45 s	
Data acquisition	Plate read		
Melting curve analysis			1
Hold	20 °C	Indefinite	1

**Table 5 antioxidants-12-00419-t005:** Items used for the telomerase reactions.

Cell lysate sample or cell lysis buffer	0.5 μL
5^x^Telomerase reaction buffer	4 μL
Nuclease-free H_2_O (Cat #8928d)	15.5 μL
Total volume	20 μL

**Table 6 antioxidants-12-00419-t006:** Items used in qPCR assay of LightCycler^®^ 480 II.

Post-telomerase reaction sample or H_2_O	1 μL
Primer stock solution (TPS)	2 μL
2^x^ qPCR master mix	10 μL
Nuclease-free H_2_O (Cat #8928d)	7 μL
Total volume	20 μL

**Table 7 antioxidants-12-00419-t007:** LightCycler^®^ 480 II qPCR assay setup.

Step	Temperature	Time	Number of Cycles
Initial denaturation	95 °C	10 min	1
Denaturation	95 °C	20 s	40
Annealing	52 °C	20 s	
Extension	72 °C	45 s	
Data acquisition	Plate read		
Melting curve analysis			1
Hold	20 °C	Indefinite	1

**Table 8 antioxidants-12-00419-t008:** Telomere length, telomerase activity and 8-OHdG levels in granulosa cells and follicular fluid in the total, pregnant and non-pregnant patients.

Parameters	Granulosa Cell	Follicular Fluid	*p*-Value
Total Patients			
**Telomere length (bp) (n = 89)**	11.7 (0–69.6)	9.9 (0–48.3)	0.991
**Telomerase activity (no dimension) (n = 90)**	0.0007 (0.0007–0.2)	0.000001 (0.0000001–0.005)	<0.001
**8-OHdG level/retrieved oocyte (ng/mL) (n = 72)**	0.2 (0.04–1.7)	2.2 (0.6–12.05)	<0.001
**Pregnant patients**			
**Telomere length (bp) (n = 24)**	12.9 (0–69.6)	7.0 (1–28.1)	0.136
**Telomerase activity (no dimension) (n = 26)**	0.0009 (0.0007–0.03)	0.000001 (0.0000001–0.005)	<0.001
**8-OHdG level/retrieved oocyte (ng/mL) (n = 22)**	0.2 (0.04–0.6)	1.7 (0.7–12.1)	<0.001
**Non-pregnant patients**			
**Telomere length (bp) (n = 65)**	10.4 (0–37.1)	10.5 (0–48.3)	0.325
**Telomerase activity (no dimension) (n = 64)**	0.0007 (0.0007–0.2)	0.000001 (0.0000001–0.003)	<0.001
**8-OHdG level/retrieved oocyte (ng/mL) (n = 50)**	0.3 (0.06–1.7)	2.6 (0.6–10.3)	<0.001

Values indicate medians and the minimum and maximum value in the parentheses. Due to non-normal distribution of the data, we compared the pairs using Wilcoxon signed-rank tests.

**Table 9 antioxidants-12-00419-t009:** Patients’ distribution according to the median values of the telomere length, telomerase activity and 8-OHdG levels.

Parameters	Non Pregnant	Pregnant	*p*-Value
Telomere length in granulosa cell (bp)			
Under median	38 (54.3%)	10 (38.5%)	0.168
Equal to or higher than median	32 (45.7%)	16 (61.5%)	
Total	70 (100%)	26 (100%)	
Telomere length in follicular fluid (bp)			
Under median	31 (45.6%)	16 (64.0%)	0.115
Equal to or higher than median	37 (54.4%)	9 (36.0%)	
Total	68 (100%)	27 (100%)	
Telomerase activity in granulosa cell (no dimension)			
Under median	34 (53.1%)	13 (50.0%)	0.788
Equal to or higher than median	30 (46.9%)	13 (50.0%)	
Total	64 (100%)	26 (100%)	
Telomerase activity in follicular fluid (no dimension)			
Under median	30 (48.3%)	16 (62.5%)	0.207
Equal to or higher than median	34 (51.6%)	10 (37.5%)	
Total	64 (100%)	26 (100%)	
8-OHdG/oocyte in granulosa cell (ng/mL)			
Under median	22 (42.0%)	11 (50.0%)	0.638
Equal to or higher than median	28 (58.0%)	11 (50.0%)	
Total	50 (100%)	22 (100%)	
8-OHdG/oocyte in follicular fluid (ng/mL)			
Under median	21 (42.0%)	15 (68.2%)	0.041
Equal to or higher than median	29 (58.0%)	7 (31.8%)	
Total	50 (100%)	22 (100%)	

To assess the possible level of association we conducted chi-square tests. The values indicate the number of observations, and percentages are in the parentheses.

**Table 10 antioxidants-12-00419-t010:** The association between indicators of IVF success and telomere length, telomerase activity and 8-OHdG levels in granulosa cells and follicular fluid.

Parameters		No. of Retrieved Oocytes	Immature Oocyte		MII Oocyte	2PN Cell (Day 1)	Cleavage Stage Embryo (Day 3)	Blastocyst (Day 5)
			GV Oocyte	MI Oocyte				
Telomere length	GC	0.09	0.05	0.01	0.07	−0.02	−0.02	0.01
	FF	−0.03	0.02	0.06	−0.04	−0.05	−0.02	−0.14
Telomerase activity	GC	0.19	0.04	0.21 *	0.09	0.04	−0.04	0.01
	FF	−0.19	−0.01	−0.06	−0.19	−0.05	0.05	−0.07
8-OHdG/no. of retrieved oocytes	GC	−0.49 **	−0.20	−0.11	−0.36 **	−0.21	−0.10	−0.11
	FF	−0.65 **	−0.20	−0.19	−0.64 **	−0.34 **	0.06	−0.32 **

** *p* < 0.01; * *p* < 0.05. Values indicate the Spearman correlation coefficients. Correlations were calculated pairwise. N = 68–102.

## Data Availability

The data that support the findings of this study are available from the corresponding author upon reasonable request.

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
