# Peer review of "Telomere Length and Telomerase Activity of Granulosa Cells and Follicular Fluid in Women Undergoing In Vitro Fertilization"

_antioxidants, 2023, doi:10.3390/antiox12020419_

Round 1

Reviewer 1 Report

The study aim is interesting and worth investigating. The article is well-described and documented. Some minor comments are provided

1. Abstract „number of 2PN cells „ please do not use abbreviations here
2. Please discuss how your methodology  manipulation of biological material might affect telomere length, telomerase activity (please see reports on growth perturbation in animals/humans after procedures prior ART is used)
3. provide a section on a study limitation section

Author Response

University of Pecs School of Medicine

Department of Obstetrics and Gynecology

Head: Kalman Kovacs, MD, Med. habil.

H-7624 Pecs, 17. Edesanyak u., Hungary

Tel: (0036)72 536 360;

Fax: (0036)72 536-000/#6360

[email protected]

Prof. Dr. Stanley Omaye

Editor-in-Chief Journal of Antioxidants

Department of Agriculture, Nutrition and Veterinary Sciences

University of Nevada

1664 North Virginia Street, Reno, NV 89557, USA,

Tel.: +1-775-784-6447

E-mail address: [email protected]

Dear Prof Dr. Omaye, and Dear Assistant Editor Ms. Renee Wei,

Please find our enclosed revised manuscript entitled, “Telomere length and telomerase activity of granulosa cells and follicular fluid in women undergoing in vitro fertilization”, which we are submitting for consideration to publish as an Original Research Article in the journal of Antioxidants.

Hereby I would like to provide a point-by-point response to each concern of the editor and the reviewers as well:

Reviewer number 1:

Thank you for your comments and critical remark, please find our answers below.

-“Abstract „number of 2PN cells „ please do not use abbreviations here.”

Thank you for your comment, according to the suggestion we omitted the abbreviation from the manuscript.

-„Revised text (Line 31)”

The text have been modified as recommended.

-“Please discuss how your methodology manipulation of biological material might affect telomere length, telomerase activity (please see reports on growth perturbation in animals/humans after procedures prior ART is used).”

Thank you for your comment, according to your suggestion we added a new paragraph to the „Discussion” section. In this part we provided information with references, how the IVF treatment itself induces reactive oxygen species (ROS) production, causing oxidative stress (OS), which accelerates telomere shortening, and decreases telomerase activity (Line 376-382).

-“Provide a section on a study limitation section.”

Thank you for your valuable suggestion, we inserted the missing study limitation section. (Line 404-413):

We have carefully proofread the text and corrected the misspellings throughout the manuscript.

After edition based on the comments independent reviewers, we wish that this original article would be considered for publication in the journal of Antioxidants. The data have not been published elsewhere, nor is this manuscript under consideration at any other journal. If I can provide any other information, please contact me by phone, fax, e-mail, or at the following address.

We thank you for reviewing our manuscript and if any additional information is needed, I will be at your disposal.

Sincerely,

Pecs, 02/05/2023.                                                                  Dr. Balint Farkas, Med. habil.

Reviewer 2 Report

Novel strategies for the management of IVF are nowadays a highly relevant topic, and any new information that could contribute to our understanding in the physiology of reproductive cells and thus to improve IVF success are welcome. In this sense, the submitted paper presents with an interesting point that could be investigated further. The study is appropriately designed and includes a good number of subjects that are well characterized, and despite the fact that the experiments could not prove the hypothesis set by the authors, it is still a valuable contribution to the field.

Nevertheless, I do have several remarks that are by and large connected to the clarity of the paper:

-          Why only 8-OHdG was selected as a marker of oxidative stress for the study? The authors should provide more feedback to oxidative damage to DNA set in the context of cell survival, particularly in IVF settings. What would be the prime sources of free radicals in the IVF environment? Overall, the authors should add a few lines to stress out the roles free radicals play in the physiology and pathophysiology of the oocyte to interconnect this aspect of the study with telomere function.

-          I would also recommend to clearly state the rationale and hypothesis for this study as well as its novelty as opposed to previous reports that may have been focused on the leading topic of this study.

-          Although the authors state that the experiment was not conclusive in the design selected for this study, they could discuss a bit what may have been the reasons for such outcome, and what could have been changed in order to confirm a direct association between telomere function and reproductive potential in women receiving IVF treatment. Also, limitations of the study should be discussed.

-          Line 141: please add more detail on the thawing, washing and centrifugation procedure as well as obtention of the cell extracts, or add a reference where the procedure has been described.

-          How was DNA purity and quantity assessed for subsequent qPCR assays?

-          Please add cities of the manufacturers from which the chemicals were purchased (where missing, for example line 97, 98 or 157).

-          Most of the equipment or machines used for the experiments are not described. What model and from what manufacturer was used for PCR or wavelength measurement during ELISA assays?

-          References in the text should be placed into square brackets.

-          Although I highly appreciate the effort, the manuscript should be carefully edited from a linguistic point of view since numerous grammar errors are found in the text. Line 45 – what is “it”? Line 47 – what are “latters”?

Author Response

University of Pecs School of Medicine

Department of Obstetrics and Gynecology

Head: Kalman Kovacs, MD, Med. habil.

H-7624 Pecs, 17. Edesanyak u., Hungary

Tel: (0036)72 536 360;

Fax: (0036)72 536-000/#6360

[email protected]

Prof. Dr. Stanley Omaye

Editor-in-Chief Journal of Antioxidants

Department of Agriculture, Nutrition and Veterinary Sciences

University of Nevada

1664 North Virginia Street, Reno, NV 89557, USA,

Tel.: +1-775-784-6447

E-mail address: [email protected]

Dear Prof Dr. Omaye, and dear Assistant Editor Ms. Renee Wei,

Please find our enclosed revised manuscript entitled, “Telomere length and telomerase activity of granulosa cells and follicular fluid in women undergoing in vitro fertilization”, which we are submitting for consideration to publish as an Original Research Article in the journal of Antioxidants.

Hereby I would like to provide a point-by-point response to each concern of the editor and the reviewers as well:

Thank you for your comments and critical remark, please find our answers below.

-“Why only 8-OHdG was selected as a marker of oxidative stress for the study? The authors should provide more feedback to oxidative damage to DNA set in the context of cell survival, particularly in IVF settings. What would be the prime sources of free radicals in the IVF environment? Overall, the authors should add a few lines to stress out the roles free radicals play in the physiology and pathophysiology of the oocyte to interconnect this aspect of the study with telomere function.”

Thank you for your comment, according to your suggestion we added two paragraphs to the „Discussion” section. In these parts we explain, why 8-OHdG is the most relevant oxidative stress marker of DNA damage in our study, what are the sources of the free radicals of an IVF procedure, and how they interfere with telomere function. We also added some new references (Line 366-382).

-“I would also recommend to clearly state the rationale and hypothesis for this study as well as its novelty as opposed to previous reports that may have been focused on the leading topic of this study.”

Thank you for your valuable recommendation. I would like to call your attention to the last paragraph of the „Introduction” section. In this part we explain, that the novelty of our study is that telomere length, telomerase activity and 8-OHdG levels were investigated simultaneously in granulosa cells (GC) and follicular fluid (FF), and our attempt was to compare the clinical values of FF and GC to analyze oxidative stress-related telomere dysfunction, because we hypothesized a clinical implication possibility in patients undergoing IVF therapy.

-“Although the authors state that the experiment was not conclusive in the design selected for this study, they could discuss a bit what may have been the reasons for such outcome, and what could have been changed in order to confirm a direct association between telomere function and reproductive potential in women receiving IVF treatment. Also, limitations of the study should be discussed.”

Following your suggestion, we created a new section „Study limitations” (Line: 404-413).

-“Line 141: please add more detail on the thawing, washing and centrifugation procedure as well as obtention of the cell extracts, or add a reference where the procedure has been described.”

According to the reviewer’s request, we added the necessary details at line 118 and line 145.

-“How was DNA purity and quantity assessed for subsequent qPCR assays?”

We added the missing information into the text. For DNA purification and quantification, we used MaestroNano Micro-Volume Spectrophotometer (Line 184).

-“Please add cities of the manufacturers from which the chemicals were purchased (where missing, for example line 97, 98 or 157).”

We completed all mentioned chemicals with cities of purchase.

-“Most of the equipment or machines used for the experiments are not described. What model and from what manufacturer was used for PCR or wavelength measurement during ELISA assays?”

The missing information was added about PCR in line 204 and 236., and about ELISA assays in line 147.

-“References in the text should be placed into square brackets.”

Thank you for note, we changed all the brackets to square one.

-“Although I highly appreciate the effort, the manuscript should be carefully edited from a linguistic point of view since numerous grammar errors are found in the text. Line 45 – what is “it”? Line 47 – what are “latters”?”

Thank you for comment, we corrected the mentioned grammar mistakes, and also revised the whole manuscript.

We have carefully proofread the text and corrected the misspellings throughout the manuscript.

After edition based on the comments independent reviewers, we wish that this original article would be considered for publication in the journal of Antioxidants. The data have not been published elsewhere, nor is this manuscript under consideration at any other journal. If I can provide any other information, please contact me by phone, fax, e-mail, or at the following address.

We thank you for your consideration of the revised manuscript, and if any additional information is needed, please contact us.

Sincerely,

Pecs, 02/05/2023.                                                                  Dr. Balint Farkas, Med. habil.
